# One Pot Self-Assembling Fe@PANI Core–Shell Nanowires for Radar Absorption Application

**DOI:** 10.3390/nano13061100

**Published:** 2023-03-19

**Authors:** Chung-Kwei Lin, Yuh-Jing Chiou, Sheng-Jung Tsou, Chih-Yi Chung, Chen-Chun Chao, Ruey-Bing Yang

**Affiliations:** 1Research Center of Digital Oral Science and Technology, College of Oral Medicine, Taipei Medical University, Taipei 110, Taiwan; 2School of Dental Technology, College of Oral Medicine, Taipei Medical University, Taipei 110, Taiwan; 3Department of Chemical Engineering and Biotechnology, Tatung University, Taipei 104, Taiwan; 4Department of Aerospace and Systems Engineering, Feng Chia University, Taichung 407, Taiwan

**Keywords:** iron nanowire, polyaniline, one pot and radar absorption

## Abstract

The one-pot process, which combines the polymerization of polyaniline (i.e., PANI) with subsequent reduction of iron nanowire (i.e., Fe NW) under a magnetic field, was developed to produce Fe@PANI core–shell nanowires. The synthesized nanowires with various PANI additions (0–30 wt.%) were characterized and used as microwave absorbers. Epoxy composites with 10 wt.% absorbers were prepared and examined using the coaxial method to reveal their microwave absorbing performance. Experimental results showed that the Fe NWs with PANI additions (0–30 wt.%) had average diameters ranging from 124.72 to 309.73 nm. As PANI addition increases, the α-Fe phase content and the grain size decrease, while the specific surface area increases. The nanowire-added composites exhibited superior microwave absorption performance with wide effective absorption bandwidths. Among them, Fe@PANI-90/10 exhibits the best overall microwave absorption performance. With a thickness of 2.3 mm, effective absorption bandwidth was the widest and reached 3.73 GHz, ranging from 9.73 to 13.46 GHz. Whereas with a thickness of 5.4 mm, Fe@PANI-90/10 reached the best reflection loss of −31.87 dB at 4.53 GHz.

## 1. Introduction

The current human lifestyle continuously increases the demand for electromagnetic (EM) energy for various devices in industrial, military, aerospace, and commercial applications [1,2,3]. While electromagnetic waves have brought tremendous development to human society, their adverse effects on the natural environment and even human health have become obvious [4,5]. Preventing the electromagnetic interference (EMI) threat has attracted increasing research and development interests concerning radar absorption materials (RAMs) [6,7,8,9,10,11,12]. Conventional iron-based microwave absorbing materials such as polycrystalline ferrites and carbonyl iron epoxy composites with appropriate complex permittivity and permeability can be used for EMI suppression. Such materials with high microwave permeability, high magnetic loss, a favorable form of frequency dependence of permeability, and a proper ratio between the permeability and permittivity (i.e., for impedance matching) exhibit superior microwave absorption performance [13,14].

Concerning permittivity, modifications of iron-based materials with conducting polymers have been attempted to improve the dielectric loss and interfacial loss. For instance, Zhang et al. [15] introduced polyaniline (PANI) to coat Fe_3_O_4_ microspheres. With 50 wt% of the obtained products used as absorbers in paraffin wax, the optimal Fe_3_O_4_/PANI core–shell structure (300 nm Fe_3_O_4_ coated with 100 nm thickness PANI) exhibited a reflection loss of −37.4 dB at 15.4 GHz. He et al. [16] synthesized PANI/carbonyl iron/Fe_3_O_4_ composite powder by mechanical mixing and the RAMs exhibited a reflection loss of −48.3 dB at 9.6 GHz. Praveena et al. [17] prepared CoFe_2_O_4_/PANI nanocomposites that exhibited a wide effective absorption at frequencies ranging from 11 to 12 and 15 to 18 GHz with a reflection loss lower than −10 dB. In addition to core–shell powder and nanocomposite, high aspect ratio nanowire materials were also used as RAMs. Yang et al. [18] synthesized Fe@Ag core–shell nanowires. With a 2:1 molar ratio of Fe:Ag, the core–shell nanowires exhibited optimal reflection loss (−58.69 dB at 7.53 GHz) at a mass fraction of 25% and a thickness of 3.36 mm, whereas Huang et al [19] synthesized coralloid NiS/Ni_3_S_4_@PPy@MoS_2_ nanowires using a three-step route. They reported that a composite with 50% fillers and a thickness of 2.29 mm exhibited the best microwave absorption performance with a reflection loss of −51.29 dB at 10.1 GHz and an effective absorption bandwidth of 3.24 GHz.

The above-mentioned reports revealed that microwave absorption performance can be improved by using conducting polymers, core–shell structures, and nanowires. However, the required sophisticated procedures may make practical applications of these nanomaterials difficult. Fe-based materials can be prepared by the reduction of Fe salts. They include thermal cracking, hydrothermal method, template-assisted development method, and borohydride sodium reduction method [20]. In addition, hydrogen gas can be used as a reducing agent for the formation of iron-based materials under relatively high temperatures [21]. In the present study, the novel one-pot process, combined with polymerization and iron nanowire reduction in a magnetic field via NaBH_4_ reduction under mild conditions [22], was applied for the synthesis of Fe@PANI nanowires to be used as radar absorption material. FeCl_3_ not only be used as the catalyst for polymerization but also as the precursor for the preparation of Fe NW. The one-pot prepared Fe@PANI core–shell nanowires were characterized, and their microwave absorption properties were determined. The development of this simple process will be beneficial to the mass production of nanosized RAMs materials.

## 2. Materials and Methods

### 2.1. Fe@PANI Core–Shell Nanowire Preparation and Characterization

The iron nanowires with PANI coating on the surface (denoted as Fe@PANI) were prepared in one pot process that includes PANI polymerization and Fe nanowire formation. PANI (5, 10, 20, and 30 wt.%) was prepared from aniline via the catalysis of FeCl_3_ in a room temperature 24 h polymerization in a batch pot [23,24]. A 95% Ar and 5% H_2_ gas mixture was used to purge the solution for 30 min to reduce the oxygen concentration and minimize the oxidation of iron nanowires. Next, sodium borohydride was added to the solution at room temperature to reduce FeCl_3_ in a parallel magnetic field [25]. The solid products were washed subsequently with deionized water, ethanol, and acetone. The products were heat treated at 200 °C for 2 h under a 95% Ar and 5% H_2_ gas mixture environment to remove the impurities. Figure 1 shows the schematic illustrations of this one-pot Fe@PANI nanowire preparation process. Table 1 summarizes the sample codes for the Fe@PANI nanowires prepared in this study.

Field emission scanning electron microscope (FE-SEM, Hitachi SU8000 Series UHR Cold-Emission Field Emission Scanning Microscope, Tokyo, Japan), X-ray diffraction (XRD, Bruker D2 PHASER X-ray Diffractometer, Billerica, MA, USA), and Fourier transform infrared spectroscopy (FTIR, Jasco FT/IR-6700 FT-IR Spectrometer, Tokyo, Japan) analytical techniques were conducted to observe the morphologies and to characterize the structures of the Fe@PANI nanomaterials, respectively. Brunauer–Emmett–Teller (BET, Porous Materials Inc. BET-201A, Ithaca, NY, USA) measurement was conducted to obtain the specific surface area of the prepared Fe@PANI nanowire. SQUID VSM test (SQUID-VSM Magnetometry—Quantum Design MPMS 3, CA, USA) was used to aid the evaluation of the magnetic behavior of the prepared nanowire under a lower magnetic field.

### 2.2. RAMs Preparation and Evaluation

The synthesized Fe@PANI core–shell nanowires (10 wt.%) were dispersed in an epoxy resin matrix to evaluate their microwave absorbing performance by coaxial method [26]. Ring-shaped specimens were prepared with a 3.04 mm inner diameter, a 7.00 mm outer diameter, and a 1.80 mm thickness. Before measurement, calibration was performed using an Agilent precision kit (85050C). The complex permittivity (ε_r_ = ε′ + iε″) and complex permeability (μ _r_ = μ′ + j μ″) were obtained using an Agilent 8510C vector network analyzer with an Agilent coaxial transmission airline (850151-60010) at frequencies ranging from 2 to 18 GHz. The reflection loss (R) of electromagnetic waves can be calculated from the measured data using the following equations [27]: (1)$$Z_{\mathrm{in}}=Z_{0}\sqrt{\frac{\mu_{r}}{\varepsilon_{r}}}\tanh\left[j\left(\frac{2\pi \mathrm{fd}}{c}\right)\sqrt{\varepsilon_{r}\mu_{r}}\right]$$
(2)$$R=20\log\left|\frac{Z_{\mathrm{in}}-Z_{0}}{Z_{\mathrm{in}}+Z_{0}}\right|$$

In these equations, Z_in_ is the input impedance when the electromagnetic wave incidence is normal to the absorber, Z_0_ is the free space impedance, f is the frequency of the electromagnetic wave, d is the thickness of the absorber, and c is the velocity of light in vacuum. The reflection loss (R) can be calculated with unit in dB.

## 3. Results and Discussion

### 3.1. Characterization of Fe and Fe@PANI Nanowires

Figure 2 shows the FE-SEM images of the PANI, Fe nanowires, and Fe@PANI nanowires with various PANI weight percentages. The PANI showed a ribbon-like microstructure with large surface area as shown in Figure 2a. The prepared Fe nanowires (Figure 2b) exhibited a typical linear structure. This is due to the assistance of the external magnetic field during the reduction of Fe ion to Fe metal. The reduced Fe nanoparticles were lined up and formed Fe nanowires. As PANI were added into the process, the Fe@PANI products with 5–30 weight percentages of PANI maintained nanowire structure. As shown in Figure 2c–f, the higher the PANI addition, the larger the diameter of the nanowire. Superfluous PANI not covering nanowires can also be observed, especially for the 30 wt.% Fe@PANI nanowires (Figure 2f)

The electron microscope images were further analyzed to determine the diameter distribution. Figure 3 shows the probability density (bar chart) and the cumulative density (line with symbols) of nanowires with different PANI weight percentages of 0, 5, 10, 20, and 30wt%. For pure Fe nanowires (0% PANI added), Figure 3a, the diameter of the nanowires ranged from 90–150 nm. After adding PANI, Figure 3b–e, the diameter distribution ranges were 160–260, 160–320, 200–340, and 260–360 nm for 5, 10, 20, and 30 wt% PANI, respectively. Figure 3f shows the average diameter with an error bar as a function of PANI concentration. The average diameters for nanowires with 0, 5, 10, 20, and 30 wt% PANI were 124.72, 207.76, 244.03, 258.11, and 309.73 nm, respectively. The greater the added PANI amount, the larger the nanowire diameter. Table 2 summarizes the average diameter, standard deviation (*s*), number of input quantities (*N*), and standard uncertainty (*u*) for nanowires, calculated according to the guide to the expression of uncertainty in measurement [28]. It can be noted that, compared to the shell-less Fe NW, the values of standard deviations and standard uncertainties increase after PANI addition.

Figure 4 shows the FE-SEM images with higher magnification for Fe, Fe@PANI-95/5, and Fe@PANI-90/10 nanowires. As shown in Figure 4a, the Fe nanowires were formed by round-shape Fe nanoparticles with a relatively smooth surface. After adding 5 and 10 wt.% PANI (Figure 4b,c), the Fe@PANI nanowires remained straight long-chain nanowires composed of round Fe nanoparticles but were now coated with PANI. It should be pointed out that with 5wt% PANI addition, Fe nanowires were not fully enveloped with PANI, as shown in the middle of Figure 4b. Fe nanowires can be fully covered by PANI with at least 10 wt.% PANI addition, Figure 4c.

Since all Fe@PANI nanowires exhibited similar X-ray diffraction patterns, Figure 5 shows the XRD patterns of pure Fe, selected Fe@PANI nanowires (10 and 30 wt.% PANI), and pure PANI. For Fe nanowires without PANI addition, shown in curve (a), there were main characteristic peaks (solid black circles) at 44.6°, 65.16° and 82.53° corresponding to α-Fe (PDF card No. 01-087-0722) in (110), (200) and (211) reflections, respectively. Fe_3_O_4_ (hollow square, PDF card No. 01-088-0315) in (220), (311), (400), and (511) reflections can also be found at 30.15°, 35.52°, 43.17°, and 57.10°, respectively. This suggests the formation of iron oxide (Fe_3_O_4_) outside the Fe nanoparticles and shows a similar trend as reported in the literature [29]. Both α-Fe and Fe_3_O_4_ structures are ferromagnetic. As more PANI were coated on Fe nanowires, curves (b) and (c) show XRD patterns more similar to that of curve (a) where both α-Fe and Fe_3_O_4_ exhibited. For 30 wt.% PANI addition, the Fe@PANI-70/30 nanowire (curve (c)) exhibited an extra broadened peak at ~20° due to the existence of PANI (curve (d)) [30].

By analyzing the XRD patterns, the phase content and grain size of α-Fe and Fe_3_O_4_ for the prepared nanowires were plotted in Figure 6, where the curves with black solid circle symbols and red hollow square symbols are α-Fe and Fe_3_O_4_, respectively. The phase content was estimated by the Rietveld fitting method [31]. One XRD pattern resulted in one estimated phase content and can only show the trend of variation. As shown in Figure 6a, the pure Fe nanowires have a phase content of 98.3 and 1.7 mol.% for α-Fe and Fe_3_O_4_, respectively. With the addition of PANI, a decrease in α-Fe phase content can be observed, accompanied by the increase in Fe_3_O_4_ phase content (red hollow square symbols shown in Figure 6a). The α-Fe phase content was 85.9 and 90.4 mol.% for 10 and 20 wt.% PANI additions, while there was a significant decrease to 51.9 mol.% for Fe@PANI-70/30 nanowires (30 wt.% PANI addition). Oxidation of Fe nanowire is inevitable. The higher the oxygen concentration, the more the Fe_3_O_4_ phase content of Fe@PANI nanowires. In addition to phase content, grain sizes were estimated according to Sherrer’s formula [32] and Figure 6b shows the variation of grain size as a function of PANI concentration. For pure Fe nanowires, the grain sizes are 25.8 and 14.1 nm for α-Fe and Fe_3_O_4_, respectively. With the addition of PANI, the grain size of α-Fe decreased significantly. With 10% PANI addition, the grain size of α-Fe decreased to 10.7 nm compared to 25.8nm for Fe nanowires. The higher the PANI addition, the smaller the grain size of α-Fe. It further decreased to 7.6 and 6.4 nm for 20 and 30 wt.% PANI addition, respectively. It is, however, interesting to note that the grain sizes of Fe_3_O_4_ (red hollow square symbols in Figure 6b) for various Fe@PANI nanowires were similar. They were 14.1, 13.7, 10.2, and 13.5 nm for 0, 10, 20, and 30 wt.% Fe@PANI nanowires.

In order to confirm the polyaniline was successfully polymerized by the Fe^3+^ oxidation method, FTIR analysis was performed before and after the polymerization and Figure 7a shows the corresponding FTIR spectra within the wavenumber ranging from 1000 to 4000 cm^−1^. Before polymerization, aniline monomer exhibited characteristic absorption peaks at 3359 cm^−1^ and 3431 cm^−1^ (assigned to amines -NH_2_ stretching vibration) and 1609 cm^−1^ and 1280 cm^−1^ (corresponded to N-H and C-N stretching vibration, respectively). After polymerization of polyaniline, N-H and C-N stretching vibration slightly shifted to 1562 cm^−1^ and ~1299 cm^−1^, respectively. The major difference can be observed at a broadened peak around 3363 cm^−1^ corresponding to amines –NH- stretching vibration [33]. Figure 7b shows a series of Fe@PANI nanowires with 0–30 wt.% PANI additions. As shown in Figure 7b, no obvious peaks can be observed for pure Fe nanowires, whereas all PANI-coated Fe nanowires exhibited a broadened peak around 3363 cm^−1^ similar to that of polyaniline. The C-N stretching vibration (~1299 cm^−1^) did not show obvious variation, whereas N-H stretching vibration (~1562 cm^−1^), due to its benzenoid unit revealed a significant change after adding sodium borohydride for FeCl_3_ reduction [33]. The transformation of the benzenoid unit into the quinoid unit can be observed and was accompanied by the appearance of a C=N characteristic peak at around 1627–1594 cm^−1^ and the disappearance of N-H stretching vibration (~1562 cm^−1^).

Since PANI exhibits a fiber-like structure, the addition of PANI may induce porous structure and increase the specific surface area and pore volume within the Fe@PANI core–shell nanowire structure. Figure 8a shows the BET results of the prepared pure Fe and Fe@PANI nanowires with different PANI concentrations. The values of specific surface area with 0, 5, 10, 20, and 30% PANI addition were 4.95, 20.34, 35.72, 50.43, and 117.07 m^2^/g, respectively. The higher the PANI added (especially with 30% PANI addition), the larger the specific surface area. The corresponding pore volume of the nanowires exhibited a similar trend. The measured total pore volumes are 0.09, 0.10, 0.12, 0.15, and 0.16 cc/g for PANI in 0, 5, 10, 20, and 30%, respectively. Peymanfar et al. have reported that the high surface area-to-volume ratio may enhance the interfacial interactions at grain boundaries and improve the microwave absorbing performance [34]. In this study, the ratio of surface area to volume value was calculated by dividing the two values of every Fe@PANI sample and they were 55.0, 203.4, 297.7, 336.2, and 731.7 m^2^/cc for PANI in 0, 5, 10, 20, and 30%, respectively. As shown in Figure 8b, a significant increase in surface area-to-volume ratio can be observed with PANI addition and the morphological effects may contribute to the microwave absorbing performance.

The low frequency (<1000 Hz) magnetic properties of the Fe and Fe@PANI-90/10 nanowires were measured via SQUID VSM at 300K with an external magnetic field of 12 KOe. Figure 9a shows the VSM results where these nanowires exhibit a typical soft magnetic hysteresis loop similar to that of α-Fe [35]. The saturation magnetization (Ms), of Fe NW (black curve), was 106.4 emu/g and decreased to 71.64 emu/g for Fe@PANI-90/10 NW. As shown in Figure 9b, the coercivity (Hc) values were 184.77 and 77.0 Oe for Fe and Fe@PANI-90/10 NWs, respectively. Recall the XRD results shown in Figure 6b, in which the grain size of α-Fe significantly decreased with PANI addition, whereas that of Fe_3_O_4_ exhibited no obvious difference. This suggests that the decrease in Ms and Hc may be mainly caused by the decrease in α-Fe grain size.

### 3.2. Microwave Absorption Performance of Fe and Fe@PANI Nanowires

It is known that microwave absorption performance generally increases with increasing amounts of added absorbers [36]. The usage of nanomaterials can reduce the relative number of added absorbers. In the present study, magnetic Fe NW was coated with conductive PANI shells. The Fe@PANI core–shell nanowires were expected to exhibit different microwave absorption performances compared to their shell-less counterpart (pristine Fe NW). 

The prepared Fe@PANI-epoxy composite rings, containing 10 wt.% Fe@PANI nanowires were examined using the coaxial method to obtain the electromagnetic parameters (i.e., complex permittivity and complex permeability) at frequencies ranging from 2 to 18 GHz. Electromagnetic parameters mainly affect the microwave absorption performance and Figure 10 shows the frequency-dependent electromagnetic parameters for Fe@PANI with different weight percentages of PANI (0–30 wt.%). The real parts of complex permittivity and complex permeability represent energy storage by the microwave absorbing material with respect to its electric and magnetic properties, respectively, whereas the imaginary parts represent the electric and magnetic energy consumptions by the absorber, respectively. It can be noted that Fe NW exhibited the largest real part of permittivity (*ε*′, Figure 10a) compared to those of Fe@PANI. Generally, the small amount of PANI addition (5%) showed the lowest ε′ value and Fe@PANI-90/10 significantly increased to the second best within the test samples. *ε*′ further decreased with the increasing amount of PANI. ε′ generally decreased with increasing test frequency ranging from 2 to 12 GHz and fluctuated within 13–18 GHz. Figure 10b shows the imaginary part of permittivity (*ε*″) where generally Fe@PANI-90/10 exhibited slightly better values when compared to other samples. The intrinsic electric dipole and interfacial polarization of the prepared RAMs can affect the dielectric properties [37]. Fe@PANI-90/10 exhibited the best permittivity performance among the prepared Fe@PANI core–shell nanowires. The permeability of the real part (*μ*′, shown in Figure 10c) tended to fluctuate close to 1 at 2–13 GHz and showed an increasing fluctuation at higher frequencies ranging from 13–18 GHz. It is interesting to note that *μ*′ of Fe@PANI-90/10 was larger than those of the other samples. Similar behavior can be observed for the imaginary part of permeability (μ″, Figure 10d). Fe@PANI -90/10 possessed better magnetic storage and magnetic loss capability than the others. The fluctuation phenomena in permittivity and permeability revealed the feature of the prepared Fe nanowires with or without PANI modification in high frequency.

Higher electric and magnetic loss may result in better microwave absorption performance. In order to compare the dielectric and magnetic dissipation of the Fe@PANI nanowires, Figure 11 shows the dielectric loss factor (tan δ_ε_ = ε″/ε′) and the magnetic loss factor (tan δ_μ_ = μ″/μ′). Generally, dielectric loss capability exhibited two major peaks at ~13 and 15 GHz, Figure 11a. Whereas the magnetic loss capability, shown in Figure 11b, shows a downward trend within ~2–14 GHz, exhibiting a peak at ~15 GHz, and another one at ~17 GHz. It can be noted that both tanδ_ε_ (Figure 11a, max. ~0.28) and tan δ_μ_ (Figure 11b, max. ~0.5) were less than one. As shown in Figure 11, this suggests that Fe@PANI-90/10 (red curve) may exhibit the best overall performance when considering both tan δ_ε_ and tan δ_μ_.

The reflection loss (R) of various Fe@PANI nanowires in different thicknesses (0–6 mm, in a step of 0.1 mm) as a function of frequency were calculated from the permittivity and permeability measurement (Figure 10) using the equations shown in Section 2.2. In order to better observe the absorption performance, only the selected curves (1–5 mm in a step of 1 mm, maximum absorption, and the one with largest effective absorption bandwidth) were shown in Figure 12. For Fe NW, Figure 12a, no effective absorption (lower than −10 dB) can be observed with a thickness of less than 2 mm. For a simulated thickness of 3 mm and above, absorption peaks at 5–9 GHz for R lower than −10 dB, i.e., more than 90% radar wave loss in the material, can be observed. The absorption peak shifted to a lower frequency with increasing thickness. For a simulated thickness larger than 4 mm, two effective absorption peaks can be observed. In general, the absorption peak at low frequency was related to the property of the absorption material, and the second absorption peak at high frequency was related to the thickness [38]. The lowest reflection loss thickness can be written according to the following equation [39,40]: (3)$$t_{m}=\frac{\mathrm{nc}}{4f_{m}\sqrt{\left|\varepsilon_{r}\right|\left|\mu_{r}\right|}}\;\left(n=1,\;3,\;5\ldots \right)$$
where t_m_ is the matching thickness of the absorber for minimum R, f_m_ is the matching frequency, c is the velocity of the light, ε_r_, and μ _r_ are the complex permittivity and complex permeability, respectively. Due to the limitation of the present vector network analyzer, only n = 1 and n = 3 are studied. 

Fe NW with a thickness of 4.6 mm (green curve) exhibited the lowest reflection loss of −35.06 dB (absorption efficiency 99.97%) at 16.13 GHz. For a thickness of 5.3 mm, it possessed a maximum effective absorption bandwidth of 2 GHz (13.6–15.6 GHz). Figure 12b shows the reflection loss of Fe@PANI–95/5 for which most of them (<15 GHz) were lower than −10 dB. It is, however, interesting to note that a composite with a thickness of 5.4 mm exhibited the best reflection loss of -43.66 dB (corresponding absorption efficiency 99.99%) at 15.87 GHz. The 5.0 mm thick sample possessed a maximum effective absorption bandwidth of 2 GHz ranging from 16 GHz to 18 GHz. As more PANI was added, Fe@PANI-90/10 NWs revealed various thicknesses with wide frequency ranges of effective absorption, Figure 12c. Composites with a thickness larger than 2 mm possessed effective absorption bandwidth and the 2.3 mm thick one exhibited a maximum effective absorption bandwidth of 3.73 GHz (9.73–13.46 GHz). For a simulated thickness larger than 4 mm, the reflection loss exhibited two effective absorption ranges, similar to that of Fe NW shown in Figure 12a. The 5.4 mm thick composite exhibited the best reflection loss of −31.87 dB (absorption efficiency 99.93%) at 4.53 GHz. After further increasing the PANI addition, Fe@PANI-80/20 and Fe@PANI-70/30, respectively, shown in Figure 12d,e exhibited a general decrease in absorption performance compared to that of Fe@PANI-90/10. For Fe@PANI-80/20, a composite with a thickness of 4.4 mm possessed the best reflection loss of −26.61 dB (absorption efficiency is 99.76%) at 16.27 GHz and the 2.0 mm thick one had the maximum effective absorption bandwidth of 2.66 GHz (12.67–15.33 GHz). Meanwhile, Fe@PANI-70/30 had the best reflection loss of −36.03 dB at 14.4 GHz with a thickness of 5.6 mm and a maximum effective absorption bandwidth ranging from 16 to 18 GHz with a thickness of 4.7 mm.

As discussed in Figure 12, it can be noted that the composite prepared by adding 10 wt.% Fe@PANI-90/10 core–shell nanowires exhibited the best overall microwave absorption performance compared to the other PANI-modified Fe NW. In order to better illustrate the absorption performance, Figure 13 shows the three-dimensional contour plots of frequency–thickness–reflection loss patterns for Fe@PANI-90/10 and its shell-less counterpart (Fe NW). 

In addition, the projection on the X-Y plane shows the relation between frequency and thickness with a reflection loss lower than −10 dB (absorption efficiency > 90%). As shown in Figure 10a, when the thickness of Fe nanowires exceeds 1.3 mm, the reflection loss starts to be lower than −10 dB (−12.26 dB at 18 GHz). As the thickness increases, the absorption peaks move to lower frequencies. For thicknesses larger than 4 mm, there are two absorption peaks at both high and low frequencies. Compared to its shell-less counterpart, Fe@PANI-90/10 exhibits superior microwave absorption performance, Figure 13b. When the thickness is 1 mm, the reflection loss at 18 GHz is −13.53 dB. The maximum reflection losses move to lower frequencies with increasing thickness. The wider green band in the projection of Figure 13b reveals that Fe@PANI-90/10 exhibits a larger maximum effective absorption band compared to Fe nanowires (Figure 13a). 

It is suggested that a high surface area to volume ratio is beneficial for microwave absorbing via interfacial interactions at grain boundaries. Recall Figure 8b, in which the surface area to volume ratio was 55 m^2^/cc for pristine iron nanowire. It increased with increasing PANI addition. It was 297.7 m^2^/cc for Fe@PANI-90/10 and reached 731.7 m^2^/cc by adding 30 wt% PANI. Superfluous PANI may contribute a lot to the high surface area to volume ratio, but limited improvement on microwave absorption. The modification of 10 wt% PANI can properly coat the Fe NW and significantly enhance the microwave absorption effect via both better dielectric and magnetic attenuation. 

## 4. Conclusions

Fe@PANI core–shell nanowires were prepared by combining the polymerization of polyaniline followed by magnetizing reduction of iron nanowires in one pot. The average diameters, the specific surface area, and the surface area to volume ratio increased with increasing PANI concentration. The average diameter was 125 nm for pristine Fe NW and increased to 244 nm for Fe@PANI-90/10, whereas the specific surface area and the surface area to volume ratio for shell-less iron NW were 4.95 m^2^/g and 55.0 m^2^/cc and, , increased to 35.72 m^2^/g and 297.7 m^2^/cc with 10% PANI addition. 

The composite rings were prepared by adding 10 wt.% Fe@PANI nanowires exhibited good microwave absorption performance. All of them exhibited wide effective absorption bandwidths ranging from 2–3.7 GHz. Among the Fe NW with or without PANI shells, Fe@PANI-90/10 exhibits the best overall microwave absorption performance with effective absorption bandwidths at broad ranges as a function of frequency and thickness.

## Figures and Tables

**Figure 1 nanomaterials-13-01100-f001:**
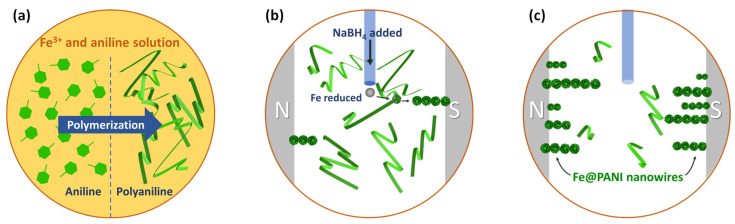
Schematic illustrations of this one-pot Fe@PANI core–shell nanowire preparation process. (**a**) Aniline monomers were polymerized into polyaniline (PANI) by Fe^3+^ ions. (**b**) Fe^3+^ ions were reduced into Fe nanoparticles with NaBH_4_ addition. Fe nanoparticles were attracted and aligned by magnetic field. (**c**) Ribbon-like PANI enveloped Fe NW to form Fe@PANI core–shell nanowires.

**Figure 2 nanomaterials-13-01100-f002:**
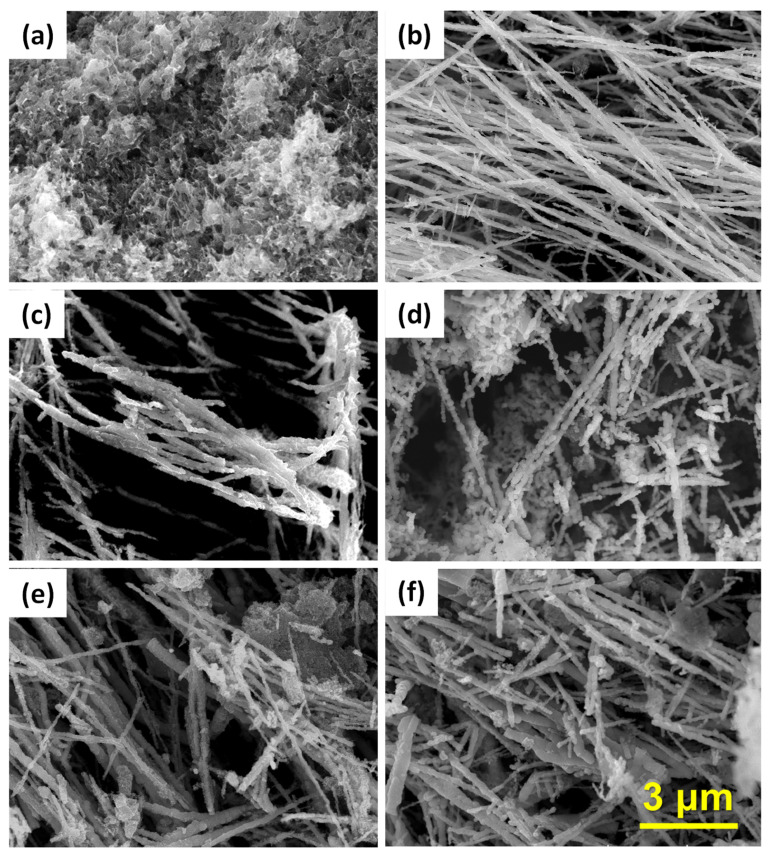
FE-SEM images of (**a**) PANI, (**b**) Fe nanowires, and Fe@PANI in different PANI weight percentages (**c**) 5wt%, (**d**) 10wt%, (**e**) 20wt%, and (**f**) 30wt%.

**Figure 3 nanomaterials-13-01100-f003:**
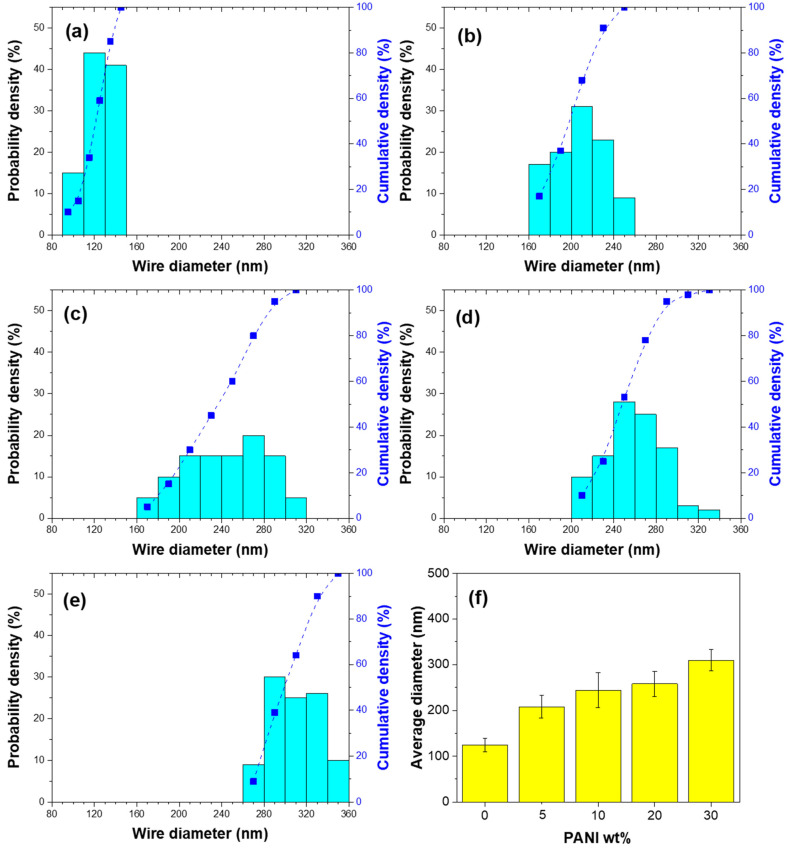
Nanowire diameter distribution, the probability density (bar chart) and the cumulative density (line with symbols), of (**a**) Fe nanowires, (**b**) Fe@PANI-95/5, (**c**) Fe@PANI-90/10, (**d**) Fe@PANI-80/20, (**e**) Fe@PANI-70/30, and (**f**) average diameter vs. PANI content. The connecting spline dotted curves are only to guide the eyes for the observation of the trend.

**Figure 4 nanomaterials-13-01100-f004:**
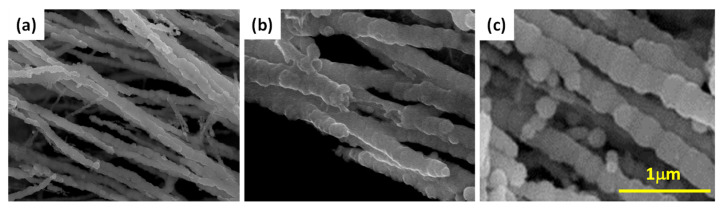
FE-SEM images of (**a**) Fe nanowires, (**b**) Fe@PANI-95/5, and (**c**) Fe@PANI-90/10.

**Figure 5 nanomaterials-13-01100-f005:**
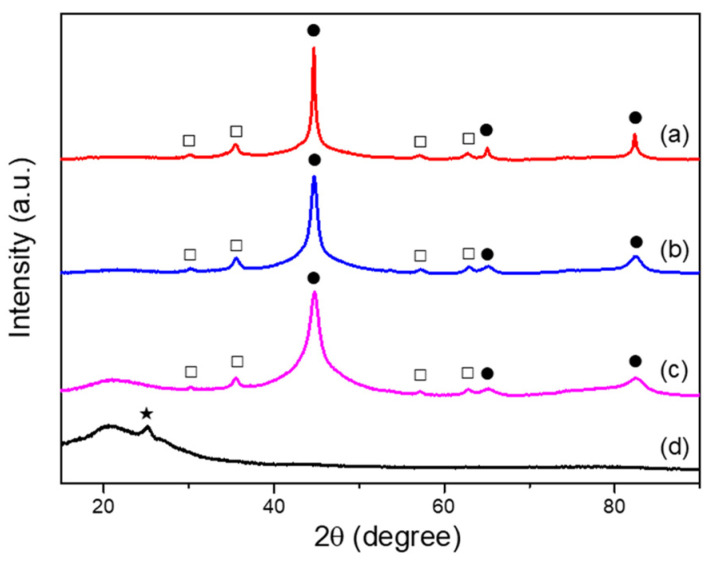
X-ray diffraction patterns of (a) Pure Fe, (b) Fe@PANI-90/10, (c) Fe@PANI-70-30, and (d) PANI. ● Fe, □ Fe_3_O_4_, ★ carbon.

**Figure 6 nanomaterials-13-01100-f006:**
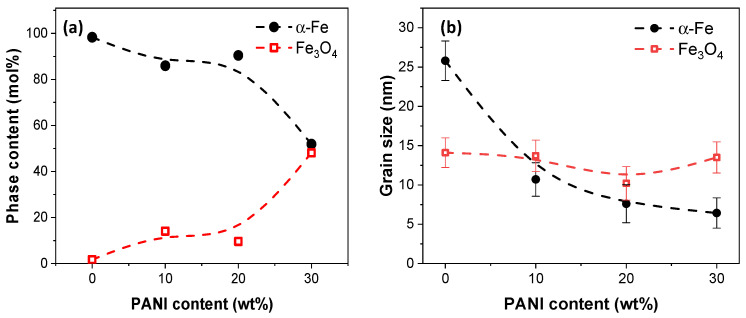
(**a**) Phase content and (**b**) grain size of α-Fe and Fe_3_O_4_ analysis via XRD. The connecting spline dotted curves are only to guide the eyes for the observation of the trend.

**Figure 7 nanomaterials-13-01100-f007:**
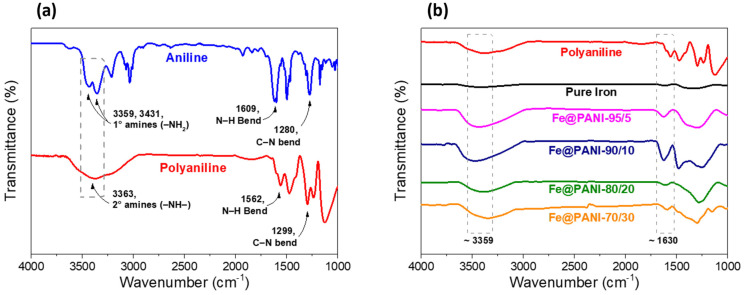
FTIR analysis (**a**) before (aniline) and after (polyaniline) polymerization, (**b**) series of Fe@PANI nanowires with 0−30 wt.% PANI additions.

**Figure 8 nanomaterials-13-01100-f008:**
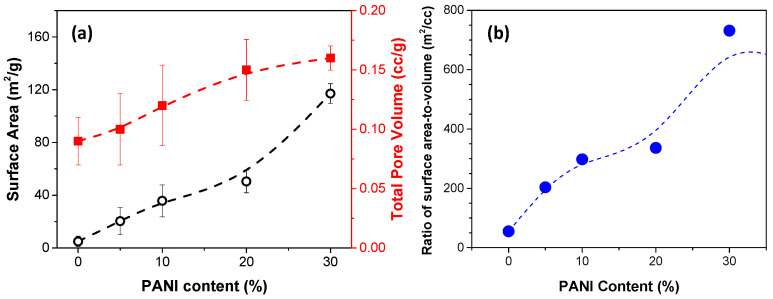
(**a**) BET specific surface area (black hollow circle) and total pore volume (red filled square), (**b**) ratio of surface area to volume (blue filled circle) as a function of PANI content. The connecting spline dotted curves are only to guide the eyes for the observation of the trend.

**Figure 9 nanomaterials-13-01100-f009:**
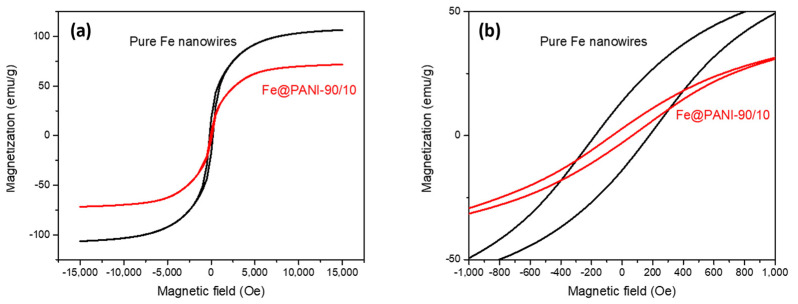
SQUID VSM results of some typical Fe@PANI nanowires.

**Figure 10 nanomaterials-13-01100-f010:**
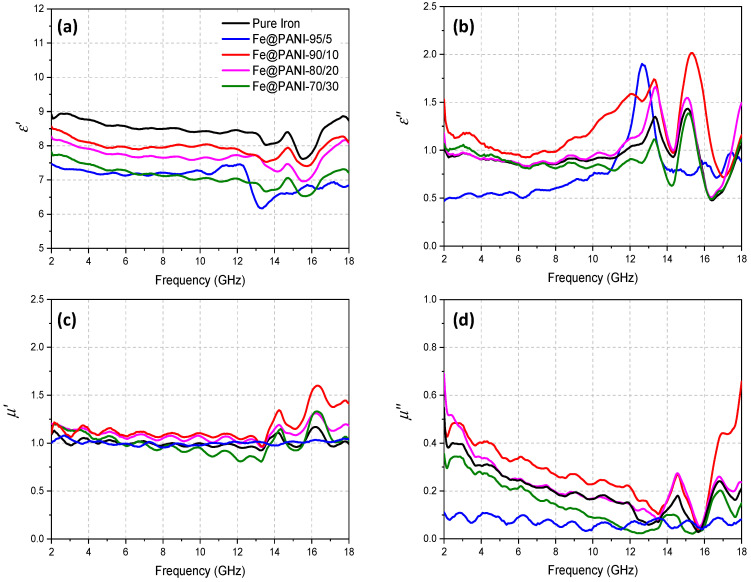
The measured dielectric (**a**) real part *ε*′, (**b**) imaginary part *ε*″, the magnetic (**c**) real part *μ*′, and (**d**) imaginary part *μ*″ of Fe@PANI nanowires composites in epoxy resin.

**Figure 11 nanomaterials-13-01100-f011:**
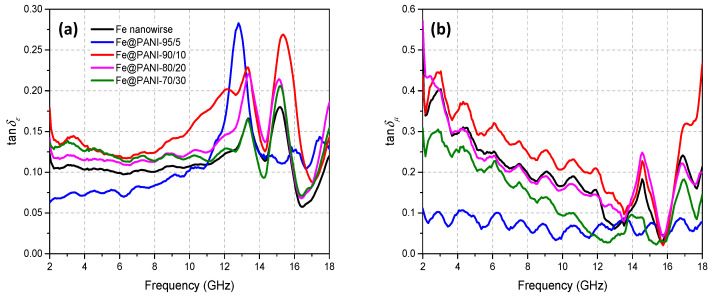
Loss tangent factors of (**a**) permittivity and (**b**) permeability.

**Figure 12 nanomaterials-13-01100-f012:**
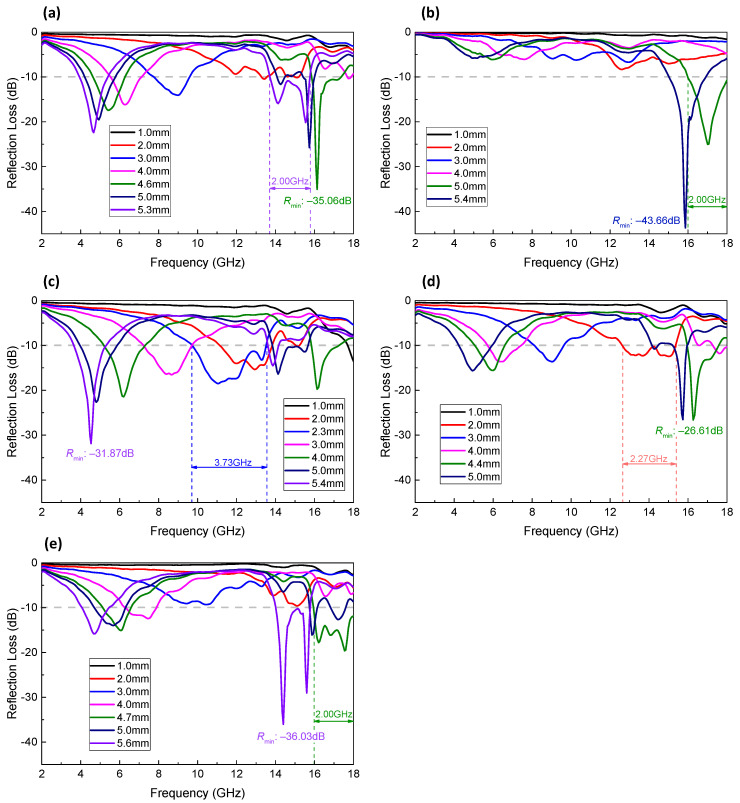
Reflection loss (R) of (**a**) Fe nanowires, (**b**) Fe@PANI-95/5, (**c**) Fe@PANI-90/10, (**d**) Fe@PANI-80/20, and (**e**) Fe@PANI-70/30 with different thicknesses.

**Figure 13 nanomaterials-13-01100-f013:**
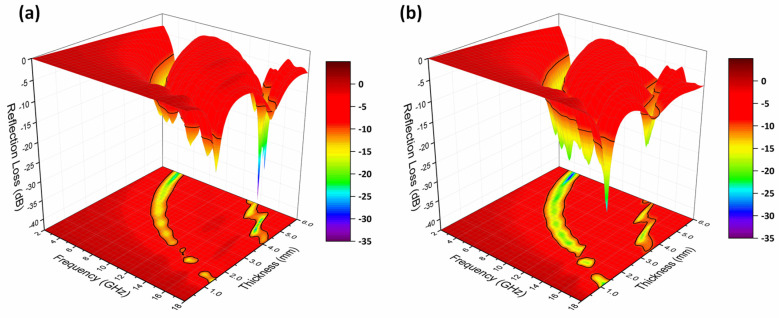
Three-dimensional representation of reflection loss of (**a**) Fe nanowires, (**b**) Fe@PANI-90/10 in different thicknesses.

**Table 1 nanomaterials-13-01100-t001:** Summary of the sample codes for the Fe@PANI nanowires.

Sample Code	Fe (wt.%)	PANI (wt.%)
Fe NW	100	0
Fe@PANI-95/5	95	5
Fe@PANI-90/10	90	10
Fe@PANI-80/20	80	20
Fe@PANI-70/30	70	30

**Table 2 nanomaterials-13-01100-t002:** Summary of average diameter, standard deviation (*s*), number of input quantities (*N*), and standard uncertainty (*u*) for nanowires.

Sample	Average Diameter (nm)	*s*	*N*	*u*
Fe NW	124.72	14.72	101	1.46
Fe@PANI-95/5	207.76	24.48	103	2.41
Fe@PANI-90/10	244.03	38.03	100	3.82
Fe@PANI-80/20	258.11	27.54	105	2.69
Fe@PANI-70/30	309.73	23.14	102	2.29

## Data Availability

Not applicable.

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
