# Peer review of "One Pot Self-Assembling Fe@PANI Core–Shell Nanowires for Radar Absorption Application"

_nanomaterials, 2023, doi:10.3390/nano13061100_

Round 1
Reviewer 1 Report
See the attachment

Reviewer 2 Report
In my opinion, the results presented are potentially highly interesting, but some control experiments are missing, and the manuscript should be improved.
Control experiments: The Fe@PANI core-shell nanowire synthesis section needs revision for other experimentalists to reproduce the results shown. Moreover, I am puzzled by the use of hydrogen gas in two separate steps during the synthesis, as H2 definitely can act as a reducing agent for metal salts. While use of hydrogen gas as a reducing agent for nanoparticle formation typically occurs at higher temperatures, the heat treatment step in this synthesis procedure is definitely within the range where one would consider H2 to be an effective reducing agent. It would be good to have at least one control sample without NaBH4 to verrify the reduction mechanism assumed in the manuscript.
In addition to this, the introduction needs a major revision. Specifically, there needs to be more emphasis on the knowledge gaps/what is not known and how this study addresses said gaps.
A minor point: I think the authors are overly optimistic regarding the average diameters listed considering the particle size distributions shown in Figure 4. Uncertainties should be added.
Reviewer 3 Report
Authors present extensive experimental results from iron nanowires reinforced by polyaniline (PANI) at various content. The experimental studies are exhaustive and the methodology is well described. The effect of PANI is well documented with direct practical applications. The content is within the scope of “nanomaterials” and could be of interest to the readers. I am mainly concerned about the authors presenting the results in a very dry format. In some cases, they should focus more on describing the mere numbers that can be extracted from the figures and explain more the reasons behind the observed trends and provide an interpretation. If they address this and the comments raised below the manuscript could be publishable in “nanomaterials”.
) In panel (c) of Fig. 1 the covering of Fe NW by PANI should be presented more clearly.
) Panel (e) of Fig. 3 (70/30 sample) seems to have an x-axis which is less zoomed compared to the rest of the panels. In parallel, for comparison purposes, given that the aim is to compare the effect of PANI inclusion on the nanowire distribution, authors could consider showing all panel using the same x-axis scale (as they do for y-axis), for example between 90 and 360 nm.
) What is the purpose of the spline connecting the content points in the panels of Figs. 6 and 8? Does it correspond to some kind of fitting? If yes, just 4 or 5 points do not justify the use of spline, unless the trend is justified. Also, both axes should have origin at zero.
) Equations should be presented better (especially Eq. 1).
) The manuscript should be checked for grammar and syntax mistakes. They are not important; still their correction would improve the clarity of the manuscript. These are related mainly to blending past/present tense plural/singular.
Line 13: “various PANI addition” -> “various PANI additions”.
Line 82: “monomer were” -> “monomers were”.
Line 122: “the 30 wt.%” -> “for the 30 wt.%”.
Line 128: “of nanowires different” -> “of nanowires with different”.
Line 150: “95/9” -> “95/5”.
Line 153: “curve a” -> “curve (a)” (same in line 160, 161 etc).
Line 175: “This … solution”, sentence is unclear and should be rephrased.
